

**Global database of actual nitrogen loss rates in coastal and marine sediments**
Yongkai Chang[1], Ehui Tan[1*], Dengzhou Gao[2], Cheng Liu[3], Zongxiao Zhang[4],
Zhixiong Huang[1], Jianan Liu[1], Yu Han[1], Zifu Xu[1], Bin Chen[5], Shuh-Ji Kao[1*]
[1] State Key Laboratory of Marine Resource Utilization in South China Sea, School of
Marine Science and Engineering, Hainan University, Haikou, China
[2] Key Laboratory for Humid Subtropical Eco-Geographical Processes of the Ministry
of Education, School of Geographical Sciences, Fujian Normal University, Fuzhou,
China
[3] Shandong Key Laboratory of Eco-Environmental Science for the Yellow River Delta,
Shandong University of Aeronautics, Binzhou, China
[4] School of Environmental Science and Engineering, Southern University of Science
and Technology, Shenzhen, Guangdong, China
[5] State Key Laboratory of Marine Environmental Science, College of Ocean and Earth
Sciences, Xiamen University, Xiamen, China
[*]**Corresponding author:**
Ehui Tan (ehuitan@hainanu.edu.cn) and Shuh-Ji Kao (sjkao@hainanu.edu.cn)





## Abstract

Denitrification and anaerobic ammonium oxidation (anammox) convert reactive nitrogen to invert $N_2$, and play vital roles in nitrogen removal in coastal and marine ecosystems, weakening the adverse effects caused by terrestrial excessive nitrogen inputs. Given the importance of denitrification and anammox in nitrogen cycle, lots of studies has measured denitrification and anammox through intact core incubations across different systems, and nitrogen loss processes are affected by a series of environmental factors such as organic carbon, nitrate, dissolved oxygen and temperature. However, a global synthesis of actual nitrogen loss rates is lacking and how environmental factors regulate nitrogen loss remains unclear. Therefore, we have compiled a database of nitrogen loss rates, including denitrification and anammox in coastal and marine systems from published literatures. This database includes 473, 466, and 255 measurements for total nitrogen loss denitrification and anammox, respectively. This work deepens our understanding of the spatial and temporal distribution of denitrification, anammox and the relative contribution of anammox to total nitrogen loss and their corresponding environmental controls. To our knowledge, the constructed database for the first time offers a comprehensive overview of actual nitrogen loss rates in coastal and marine ecosystems on a global scale. This database can be utilized to compare nitrogen loss rates of different regions, identify the key factors regulating these rates, and parameterize biogeochemical models in the future. This database is available in Figshare repository :
https://doi.org/10.6084/m9.figshare.27745770.v3 (Chang et al., 2024).



40 **KEYWORDS:** nitrogen cycle, denitrification, anammox, coastal and marine

41 ecosystems, isotope pairing technology, intact core incubations



## 1 Introduction

The production of anthropogenic reactive nitrogen has intensified remarkably since the mid-20th century to meet the increasing global population (Kennedy, 2021). It is estimated that nitrogen is entering Earth's ecosystems at more than twice its natural rate, drastically disrupting the pristine nitrogen cycle (Canfield et al., 2010). Much of the excess nitrogen, primarily in the form of nitrate, is conveyed downriver to coastal and marine systems due to the low use efficiency of crops (Cui et al., 2013), resulting in a series of environmental issues including harmful algal blooms, eutrophication, and hypoxia (Dai et al., 2023). Consequently, it is critical to understand the transformations, particularly the fates of reactive nitrogen, encountering the fact that the nitrogen cycle has been intensively altered and is currently functioning beyond the safe operating space for humanity (Richardson et al., 2023).

Denitrification and anammox are two key nitrogen loss processes in aquatic environments, playing important roles in mitigating the adverse effects of excessive nitrogen inputs (Chen et al., 2021; Tan et al., 2022). Denitrification is the sequential reduction of nitrate, nitrite, nitric oxide, and nitrous oxide ($N_2O$) to dinitrogen gas ($N_2$), which is the most energetically favorable respiratory pathway in the absence of oxygen (Devol, 2015), serving as the predominant mechanism for nitrogen loss in coastal ecosystems (Damashek & Francis, 2018; Deng et al., 2024). Anaerobic ammonium oxidation (Anammox), an alternate nitrogen loss pathway, utilizes nitrite and ammonium to generate $N_2$ with no greenhouse gas $N_2O$ production under anaerobic conditions (Graaf et al., 1995), and is a chemoautotrophic process with no



direct demand for organic carbon (Strous et al., 1999). Therefore, anammox is an
environment-friendly and energy-saving process compared to denitrification.
The $^{15}$N isotope pairing technique (IPT) has been applied to a variety of sediments to
quantify nitrogen loss rates in these settings (Nielsen, 1992; Robertson et al., 2019).
Slurry incubation and intact core incubations in combination with IPT are two widely
used methods for studying benthic nitrogen transformation pathways (Song et al.,
2016b). Slurry incubations have been used to estimate the potential rates due to
advantage of simple operation in incubations (Thamdrup & Dalsgaard, 2002), and a
large number of studies have used this method to study sediment nitrogen loss.
However, slurry incubations could not reflect the genuine benthic nitrogen
transformation rates, as the natural gradients of substrates and redox in sediments
were disrupted during incubations (Trimmer et al., 2006). The application of intact
core incubations can overcome this drawback and will enable us to fully clarify and
understand the nitrogen cycle in field aquatic ecosystems.
Over the past thirty years, the introduction of isotope pairing technology has enabled
numerous studies to measure anammox and denitrification using intact core
incubations across a range of coastal and marine environments. These environments
include intertidal wetlands (Adame et al., 2019; Liu et al., 2020), estuaries and coasts
(Chen et al., 2021; Cheung et al., 2024; Deek et al., 2013; Hellemann et al., 2017),
lagoons (Bernard et al., 2015; Magri et al., 2020) and oceans (Deutsch et al., 2010; Na
et al., 2018). Despite decades of research, the majority of studies on denitrification
and anammox have been limited to local or regional scales. Various environmental



factors, such as the availability of organic carbon (Yin et al., 2015) and nitrate
(Asmala et al., 2017), dissolved oxygen (Bonaglia et al., 2013; Song et al., 2021), and
temperature (Tan et al., 2022) influence these processes in coastal marine ecosystems.
However, to date, the global patterns and drivers of sediment nitrogen loss rates
remain poorly understood in coastal and marine systems.
In view of the critical role of nitrogen removal processes and the current lack of a
comprehensive database on actual nitrogen loss in coastal and marine systems, we
have integrated actual nitrogen loss rates, including denitrification and anammox,
from published studies, and constructed a dataset on nitrogen removal rates in these
systems. This study provides a global-scale overview of the biogeography and
potential controlling factors of denitrification and anammox in coastal and marine
ecosystems. It also highlights the potential applications of this database such as using
machine learning to predict the distribution of denitrification and anammox and
offering a crucial dataset for the parameterization and development of biogeochemical
models.
**2 Methods**
**2.1 Data compilation**
Nitrogen loss rates, including denitrification and anammox measured through intact
core incubations in coastal and marine ecosystems, were extracted from the literature
published between 1996 and 2024. Table 1 summarized the locations, observation
numbers, core incubation methods and references of nitrogen loss rates measurements.



The intact core incubations in this study include both traditional core incubations
(Bonaglia et al., 2017; Cheung et al., 2024) and continuous flow experiments
combined with core incubations (Liu et al., 2020; McTigue et al., 2016). The
peer-reviewed articles compiled in this study were sourced from the Web of Science
database as of June 2024. The search terms were "denitrification" or "anammox" or
"nitrogen loss" or "nitrogen removal". Only data where denitrification and/or
anammox rates were measured using intact core incubations combined with $^{15}$N
isotope pairing techniques were included, while those measured via slurry incubation
were excluded. The intact core incubation experiments were primarily conducted in
dark conditions and near-*in situ* or *in situ* ambient temperatures. In cases where
nitrogen loss rates were measured under both light and dark conditions, only those
measured in the dark were included. Measurements under light conditions have been
detailed in studies reported by Bartoli et al. (2021), Chen et al. (2021),
Risgaard-Petersen et al. (2004), Rysgaard et al. (1996b), and Welsh et al. (2000).
Some studies have investigated the changes in nitrogen loss processes under varying
oxygen concentrations (Bonaglia et al., 2013; Neubacher et al., 2011; Song et al.,
2021), however, only nitrogen loss rates measured under ambient oxygen
concentrations were extracted for this database. Additionally, studies examining the
effects of meiofauna or antibiotics on nitrogen removal (Bonaglia et al., 2014b; Wan
et al., 2023) were not included, only rates measured without meiofauna or antibiotic
additions were considered. At least one environmental variable was recorded for each
selected study, and means and sample sizes had to be reported for nitrogen removal



rates. Articles that only reported nitrogen loss rates without any environmental
variables were excluded. Data on total nitrogen loss rates (the sum of denitrification
and anammox), denitrification rates, anammox rates, and related environmental
variables were collected from tables, text, and/or supplementary materials, and in
some cases, extracted from graphs using Origin 2020 software. The unit conversions
were performed where necessary. In addition, longitude and latitude were extracted
from figures from published articles if not shown in the main text.

The database includes observation details (year of sampling, month of sampling,
latitude, and longitude), sediment parameters, and water physicochemical factors,
such as sediment organic carbon, the ratios of carbon to nitrogen (C/N ratios), oxygen
penetration depth, and water salinity, depth, temperature, DO, ammonium and nitrate
concentrations. Note that some environmental variables were not reported in the
original studies. NM represents parameters that were not measured, and empty or NA
indicates data not available or reported. In total, the database comprises 473, 466, 255,
and 255 measurements of total nitrogen loss rates, denitrification rates, anammox
rates, and the relative contribution of anammox to total nitrogen loss, respectively.
Authors and interested readers are welcomed to contact us to indicate an error or
update the data in the database.
For quality control, extreme nitrogen loss rate values were excluded from the database
following Chauvenet's criterion (Glover et al., 2011), a method typically applied to
normally distributed data to identify outliers whose deviation from the mean has a





probability lower than 1/(2n). More details about Chauvenet's criterion can be found
in Glover et al., (2011) and Buitenhuis et al. (2013). Very high rates of denitrification
were observed in the Tama Estuary, Japan (Usui et al., 2001), a constructed wetland in
Casino, NSW, Australia (Erler et al., 2008), a coastal lagoon in Sacca di Goro lagoon,
Italy (Magri et al., 2020) and the Tropical Coastal Wetlands, Australia (Adame et al.,
2019). For anammox, high rates were found only in a constructed wetland in Casino,
NSW, Australia (Erler et al., 2008). Similarly, high values for anammox's contribution
to total nitrogen loss were observed in the Changjiang River Estuary, China (Liu et al.,
2020), the Norwegian Trench, Skagerrak (Trimmer et al., 2013), and the Great Barrier
Reef lagoon (Erler et al., 2013), with contributions exceeding 70%. Observations with
nitrogen loss rates of 0 or NA were excluded from the outlier analysis. For example,
anammox rates of 0 were reported in the Changjiang River Estuary, China (Liu et al.,
2020), the North Sea (Neubacher et al., 2011; Rosales Villa et al., 2019), the Pearl
River Estuary, China (Tan et al., 2019), the Norwegian Trench, Skagerrak (Trimmer et
al., 2013), and the Gulf of Finland, Baltic Sea (Jäntti et al., 2011). After excluding
observations of 0 and NA (0, 8, 252, and 253 observations for total nitrogen loss rates,
denitrification rates, anammox rates, and anammox's contribution to total nitrogen
loss), the nitrogen loss rates were natural-log transformed for further analysis.

## 170 2.2 Methods for measuring denitrification and anammox rates

Before the discovery of anammox, denitrification was regarded as the sole significant
pathway responsible for nitrogen loss (Dalsgaard & Thamdrup, 2002). The $^{15}N$





isotope pairing technique (IPT) was developed to quantify denitrification rates
(Nielsen, 1992). In this method, the overlying water of intact sediment cores is
enriched with $^{15}NO_3^-$, which is mixed with the naturally occurring $^{14}NO_3^-$. After a few
hours of incubation, the denitrification products, $^{15}N$-labeled dinitrogen gas ($^{29}N_2$ and
$^{30}N_2$), are measured. Incubations to measure nitrogen loss rares have been mostly
conducted in dark conditions and near-*in situ* or *in situ* ambient temperatures. After
incubating for 1 h to over 96 h, the incubation is halted by injecting saturated $HgCl_2$ or
$ZnCl_2$ saturation solution or 37% formaldehyde. The samples are then preserved for
$^{15}N_2$ gas analyses through isotope ratio mass spectrometer (IRMS) or membrane inlet
mass spectrometry (MIMS). Key experimental details, such as incubation conditions,
temperature control, incubation time, termination, and calculation references, are
compiled in the database if provided in the original studies. For more detailed
experimental information, refer to the corresponding references.
The production rate of unlabeled $^{14}NO_3^-$ (IPT$p14$, also referred to as the genuine
production of $N_2$) can be calculated based on the assumption of random isotope
pairing during the denitrification of the uniformly mixed $NO_3^-$ species. The following
equation is commonly used to estimate the genuine $N_2$ production (Nielsen, 1992;
Steingruber et al., 2001).
$$\mathrm{IPT}p14 = \frac{p^{29}\mathrm{N}_2}{2 \times p^{30}\mathrm{N}_2} \times (p^{29}\mathrm{N}_2 + 2 \times p^{30}\mathrm{N}_2) \qquad (1)$$

Where $p^{29}N_2$ and $p^{30}N_2$ represent the total production rates of $^{29}N_2$ and $p^{30}N_2$,
respectively.
Thamdrup and Dalsgaard (2002) were the first to quantify anammox through



anaerobic slurry incubations in natural environments, discovering that anammox
could account for more than 60% of total $N_2$ production. This highlighted the
significant role of anammox in nitrogen removal. Following this, Risgaard-Petersen et
al. (2003) proposed a modification to the traditional IPT, allowing for more accurate
quantification of true $N_2$ production in environments where anammox and
denitrification coexist. This revision also enables the distinction between N2 produced
by anammox and denitrification. The revised IPT (rIPT) follows the same procedure
as the classical IPT, with $^{15}NO_3^-$ added to the overlying water of intact sediment cores,
though the calculation process is more complex. The following equations are
commonly used to estimate the actual $N_2$ production (rIPT$p$14) and denitrification
($p$14DEN) as well as anammox ($p$14ANA) (Risgaard-Petersen et al., 2003; Trimmer
& Nicholls, 2009; Trimmer et al., 2006). The total $N_2$ production rate is the sum of
denitrification and anammox rates.
$$\text{rIPT}p14 = 2r_{14} \times (p^{29}N_2 + p^{30}N_2 \times (1 - r_{14})) \tag{2}$$
$$p14\text{DEN} = 2r_{14} \times (r_{14} + 1) \times p^{30}N_2 \tag{3}$$
$$p14\text{ANA} = 2r_{14} \times (p^{29}N_2 - 2 \times r_{14} \times p^{30}N_2) \tag{4}$$
In these equations, $p^{29}N_2$ and $p^{30}N_2$ are the total production rates of $^{29}N_2$ and $p^{30}N_2$,
respectively, and $r_{14}$ represents the ratio of $^{14}NO_3^-$ and $^{15}NO_3^-$ in the nitrate reduction
zone. There are 3 different methods to estimate $r_{14}$, with detailed explanations
available in Trimmer et al. (2006).
Subsequently, Hsu and Kao (2013) revised the rIPT method to incorporate both $N_2O$
production and anammox, enabling the determination of the absolute rate of each



nitrogen loss pathway, including denitrification, anammox, and $N_2O$ production from
denitrification. Denitrification and anammox measurements based on the method of
Hsu and Kao (2013) are included in this database, whereas data on the true $N_2O$
production rate has not been included.

## 3 Results and discussion

### 3.1 Overview of the database

Overall, there are 473, 466, and 255 measurements for total nitrogen loss
denitrification and anammox, respectively (Fig. 1). Denitrification and anammox have
been measured simultaneously at 255 observations. The observations of nitrogen loss
rates are primarily distributed in the Eastern coast of the United States, the Baltic Sea,
the Eastern Coast of China, the Eastern Coast of Australia, and polar regions of the
Northern Hemisphere (Fig. 1a). Before 2000, nitrogen loss measurements were
predominantly focused on denitrification, while both denitrification and anammox
rates have been measured concurrently since 2000 (Fig. 1b). Notably, more
observations were recorded in 2011 and 2017. The studies in 2011 were mainly
conducted in the Changjiang estuary and its adjacent East China Sea (Song et al.,
2021), the Jinpu Bay, China (Yin et al., 2015), the North Sea (Bale et al., 2014), the
Northern Baltic Proper (Bonaglia et al., 2014a) and the hypoxic zone off the
Changjiang River estuary, China (Yang et al., 2022). In 2017, high observations were
found in the Northern East China Sea, China (Chang et al., 2021), the Changjiang
River Estuary, China (Liu et al., 2020; Liu et al., 2019; Tan et al., 2022), the Coast of





Victoria, Australia (Kessler et al., 2018) and the Jiulong River Estuary, China (Tan et
al., 2022).

## 3.2 Distribution of denitrification

In total, the vast majority of nitrogen loss rate measurements were conducted in the
Northern Hemisphere, and data in the Southern Hemisphere were limited (Fig. 2a, 2b,
2c). The low and middle latitudes of the Northern Hemisphere have a large body of
observations, especially in the 20-30°N, 30-40°N, and 50-60°N latitude bands.
Denitrification rates ranged from 0.04 to 750 $\mu$mol N m$^{-2}$ h$^{-1}$, with a median value of
7.72±4.30 $\mu$mol N m$^{-2}$ h$^{-1}$. There is a decreasing trend in the denitrification rates with
latitude in the Northern Hemisphere, though the observations in the high latitude are
still limited. The measurements of denitrification were mostly conducted in later
spring, summer, and early autumn, from April to September (Fig. 2d, 2e, 2f). On a
global scale, no clear seasonal pattern for denitrification rates was observed.

## 3.3 Distribution of anammox

From a latitude perspective, the distribution of anammox rates closely mirrored that of
denitrification, with the majority of observations concentrated in the 20-30°N,
30-40°N, and 50-60°N latitude bands (Fig. 3a, 3b, 3c). However, compared to
denitrification, there were fewer anammox observations. Anammox rates spanned



from 0.01 to 48.94 µmol N m$^{-2}$ h$^{-1}$, with a median value of 1.00±0.39 µmol N m$^{-2}$ h$^{-1}$.
Similar to denitrification, anammox rates also showed a decreasing trend with
increasing latitude in the Northern Hemisphere. Numerous anammox measurements
were conducted between April and September, consistent with the timing of
denitrification measurements (Fig. 3d, 3e, 3f). Additionally, February saw a high
number of anammox observations, and these observations were predominantly
conducted at the north East China Sea (Chang et al., 2021), the Changjiang Estuary
(Liu et al., 2019) and the Northeastern New Zealand continental shelf regions
(Cheung et al., 2024). On a global scale, there was no clear seasonal pattern for
anammox rates.

## 3.4 Distribution of contribution of anammox to total N$_2$ production

The relative importance of anammox to total N$_2$ production increased first and then
decreased, peaking in the 40-50°N latitudinal band in the Northern Hemisphere,
although data points in this band were limited (Fig. 4). The contribution of anammox
to total N$_2$ production varied from 0.22% to 67.33%, with a median value of 12.29%.
The highest value (67.33%) was recorded at a site on the North Atlantic continental
slope at a depth of 2000 m (Trimmer & Nicholls, 2009), where anammox accounted
for the majority of nitrogen removal. There were no significant monthly changes in
the relative importance of anammox to total nitrogen loss, except for March, when



anammox contributed a notably high percentage. High values in March were observed
in the Ulleung Basin, East Sea, and the continental shelf and slope, North Atlantic (Na
et al., 2018; Trimmer & Nicholls, 2009) where the stations were characterized by low
nitrate levels or deep water. These environmental conditions may inhibit
denitrification, thereby increasing the relative contribution of anammox to nitrogen
loss.

## 3.5 Control factors on nitrogen loss rates

The variations in denitrification rates, anammox rates, and the contribution of
anammox to total $N_2$ production (%) were compared against several environmental
variables, including sediment organic carbon, the ratios of carbon to nitrogen (C/N
ratios) and oxygen penetration depth, and water depth, temperature, salinity, dissolved
oxygen, ammonium, and nitrate concentrations. This comparison was conducted to
evaluate the controlling factors of nitrogen loss rates and the relative importance of
anammox to total nitrogen removal.
There was no significant relationship between denitrification rates and the contents of
sediment organic carbon ($p>0.05$; Fig. 5a). Heterotrophic denitrification is primarily
carried out by facultative anaerobic heterotrophs (Devol, 2015), which use organic
carbon as an electron donor and energy source. Therefore, higher organic carbon
levels might be expected to promote denitrification (Damashek & Francis, 2018).
However, no such relationship was observed in this dataset. Denitrification rates



increased with sediment carbon nitrogen ratios ($r$=0.23, $p$<0.01; Fig. 5b). The C/N
ratios can indicate the reactivity of sediment organic material, with lower C/N values
generally representing more reactive organic matter (Cheung et al., 2024; Erler et al.,
2013). Typically, high denitrification rates are associated with sediments that have
lower C/N ratios. However, in this analysis, the opposite trend was observed. One
possible explanation is that microbial communities may adapt to use organic matter
typically encountered, though the organic matter is not labile (Salk et al., 2017).
Denitrification rates showed a weak negative correlation with oxygen penetration
depth ($r$=-0.29, $p$<0.01; Fig. 5c), as greater $O_2$ penetration may be adverse to the
occurrence of denitrification (Cheung et al., 2024). Denitrification rates also
decreased with water depth ($r$=-0.2, $p$<0.01; Fig. 5d), with most observations
occurring at depths shallower than 250 m. Denitrification was positively correlated
with higher water temperatures ($r$=0.38, $p$<0.01; Fig. 5e), and negatively correlated
with salinity ($r$=-0.15, $p$<0.01; Fig. 5f), with most rates falling within two salinity
ranges (0-10 and 30-40). Samples that had a salinity greater than 40 were collected in
hypersaline lagoons of tropical regions (Enrich-Prast et al., 2016). The relationship
between denitrification and salinity across coastal environments has been summarized
by Torregrosa-Crespo et al. (2023) and will not be further elaborated here. There was
a weak negative relationship between denitrification rates and dissolved oxygen
concentrations ($r$=-0.23, $p$<0.01; Fig. 5g). Overall, higher denitrification rates were
recorded in areas with high nitrate concentrations ($r$=0.16, $p$<0.01; Fig. 5h),
suggesting the importance of nitrate substrate in regulating denitrification, though





some high rates were also observed in sites with low nitrate levels. No significant
correlation was found between denitrification rates and ammonium concentrations
($p$>0.05; Fig. 5i).

Anammox rates showed a weak positive correlation with sediment organic carbon
($r$=0.16, $p$<0.05; Fig. 6a). Although anammox is an autotrophic process that does not
require organic carbon as an electron donor (Salk et al., 2017), some studies have
reported links between sediment organic carbon content and anammox rates. For
example, studies in subtropical mangrove sediments (Meyer et al., 2005) and the
Thames estuary (Trimmer et al., 2003) found that higher organic matter stimulated
anammox. This correlation may be due to enhanced mineralization leading to
increased ammonium production, which indirectly stimulates anammox (Damashek &
Francis, 2018), as sediment organic carbon can serve as a proxy for organic carbon
mineralization (Song et al., 2016a). Similar to denitrification, high anammox rates
were observed at sites with elevated C/N ratios ($r$=0.33, $p$<0.01; Fig. 6b). More
research is needed to reveal the influencing mechanisms of organic matter quantity
and quality on anammox. No clear trend was found between anammox rates and
oxygen penetration depth ($p$>0.05; Fig. 6c), and high anammox rates were observed in
shallow waters ($p$>0.05; Fig. 6d). Anammox rates showed a weak positive correlation
with temperature ($r$=0.19, $p$<0.01; Fig. 6e). While several studies have suggested that
low temperatures could favor anammox (Dalsgaard & Thamdrup, 2002; Rysgaard et
al., 2004; Tan et al., 2020), these studies primarily measured anammox potential using





anaerobic slurry incubations. Contrary to previous findings, our study showed that
actual anammox rates increased with rising temperatures, suggesting a discrepancy
between the effects of temperature on actual and potential anammox rates. Future
research is needed to investigate the underlying mechanisms for these inconsistent
results. Anammox rates decreased with increasing salinity ($r$=-0.38, $p<0.01$; Fig. 6f),
and showed no significant relationship with dissolved oxygen ($p>0.05$; Fig. 6g). A
weak positive correlation was observed between anammox rates and nitrate
concentration ($r$=0.41, $p<0.01$; Fig. 6h), highlighting the importance of substrates in
regulating anammox. Although anammox uses nitrite as an electron acceptor rather
than nitrate (Graaf et al., 1995), nitrate reduction can produce nitrite, which promotes
anammox activity. No relationship was found between anammox rates and
ammonium concentration ($p>0.05$; Fig. 6i).

Numerous studies have found that denitrification was linked to anammox in different
habitats, including estuary sediments (Liu et al., 2020), coastal wetland sediments
(Gao et al., 2017) and paddy soils (Shan et al., 2016). Consistent with their findings,
this work also found denitrification was positively correlated to anammox ($r$=0.67,
$p<0.01$; Fig. 7). A majority of denitrifying bacteria are heterotrophic and capable of
utilizing organic matter, and the decomposition of organic matter is accompanied by
the production of ammonium (Devol, 2015), supplying substrates for anammox. Thus,
the relationship between denitrification and anammox may suggest the tight coupling
of these two nitrogen removal pathways.




There was a positive correlation between the contribution of anammox to total $N_2$
production (ra) and water depth ($r$=0.59, $p$<0.01; Fig. 8d). Previous studies have
reported similar findings, including those conducted on the Northeastern New
Zealand continental shelf (Cheung et al., 2024), the continental shelf and slope, North
Atlantic (Trimmer & Nicholls, 2009). The increased importance of anammox can be
attributed to the significant attenuation of denitrification with depth, as the availability
of organic carbon, which is essential for denitrification, decreases with increasing
water depth (Thamdrup, 2012). In addition to water depth, other factors such as
oxygen penetration depth, C/N ratios, and temperature may also influence the relative
importance of anammox. The contribution of anammox to total $N_2$ production (ra)
was positively correlated with oxygen penetration depth ($r$=0.7, $p$<0.01; Fig. 8c). As
previously mentioned, denitrification decreases with higher oxygen penetration depth,
likely increasing the relative importance of anammox indirectly. Conversely, ra
showed a decreasing trend with elevated C/N ratios ($r$=-0.35, $p$<0.01; Fig. 8b). High
C/N ratios may promote denitrification more significantly than anammox because
both processes tend to enhance with increasing C/N ratios, leading to a decrease in the
relative importance of anammox at sites with high C/N ratios. Additionally, ra was
negatively correlated with temperature ($r$=-0.29, $p$<0.01; Fig. 8e), indicating that
denitrification is stimulated at higher temperatures compared to anammox.
Temperature-controlled experiments have confirmed that denitrification has a greater
optimal temperature than anammox (Canion et al., 2014; Tan et al., 2020). No



correlations were found between ra and other environmental factors, including
sediment organic carbon, water salinity, dissolved oxygen, nitrate, and ammonium
concentrations. (all $p > 0.05$; Fig. 8a, 8f, 8g, 8h, 8i).

## 4 Applications of the database

This database serves as a valuable resource for the broad scientific communities that
are interested in nitrogen cycle processes within coastal and marine ecosystems,
particularly those focusing on denitrification and anammox. The data is made
accessible as a basic database that will lead to a deeper understanding and generate
new scientific insights into the nitrogen cycles at the global scale. Potential
applications of this database include: (1) serving as a reference for comparing
denitrification and anammox rates across local, regional, and global scales in future
studies; (2) identifying and comparing the controlling factors of denitrification and
anammox at various spatial scales; (3) predicting the global biogeography of
denitrification and anammox in coastal and marine systems through machine learning;
and (4) providing essential data for the parameterization, validation and enhancement
of Earth system biogeochemical models.

## 5 Conclusions

We compiled a global database of denitrification and anammox measurements
obtained from core incubation experiments in coastal and marine sediments. To our



knowledge, no efforts have been made to compile actual nitrogen loss rates and
associated environmental factors in coastal and marine regions on a global scale. This
database offers valuable insights into the spatiotemporal variations and potential
controlling factors of denitrification and anammox, along with the contribution of
anammox to total $N_2$ production. It can be used to compare these two nitrogen loss
processes, assess the environmental controls on them at regional and global levels,
and support the parameterization and development of biogeochemical models.

## Data availability

The data used in this study are openly available in Figshare repository at
https://doi.org/10.6084/m9.figshare.27745770.v3 (Chang et al., 2024).

## Author contributions

SJK and EHT conceived the research. YKC and EHT compiled the data. YKC, EHT,
DZG, CL and SJK participated in the data analysis. All co-authors contributed to the
writing and reviewing of this manuscript.

## Competing interests

None declared.

## Acknowledgements

We thank the authors for their contributions to the data used in this database. Thanks
to the editors and reviewers for their constructive comments and suggestions that

 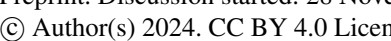 

improved this manuscript greatly.

## Financial support

This work was supported by the National Natural Science Foundation of China
(92251306 and 42276043), the Hainan Provincial Natural Science Foundation of
China (623RC456), the Collaborative Innovation Center of Marine Science and
Technology in Hainan University (XTCX2022HYC19), the Innovational Fund for
Scientific and Technological Personnel of Hainan Province (KJRC2023B04) and the
Shandong Provincial Natural Science Foundation of China (ZR2023QD103).

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

**Figures and Table**

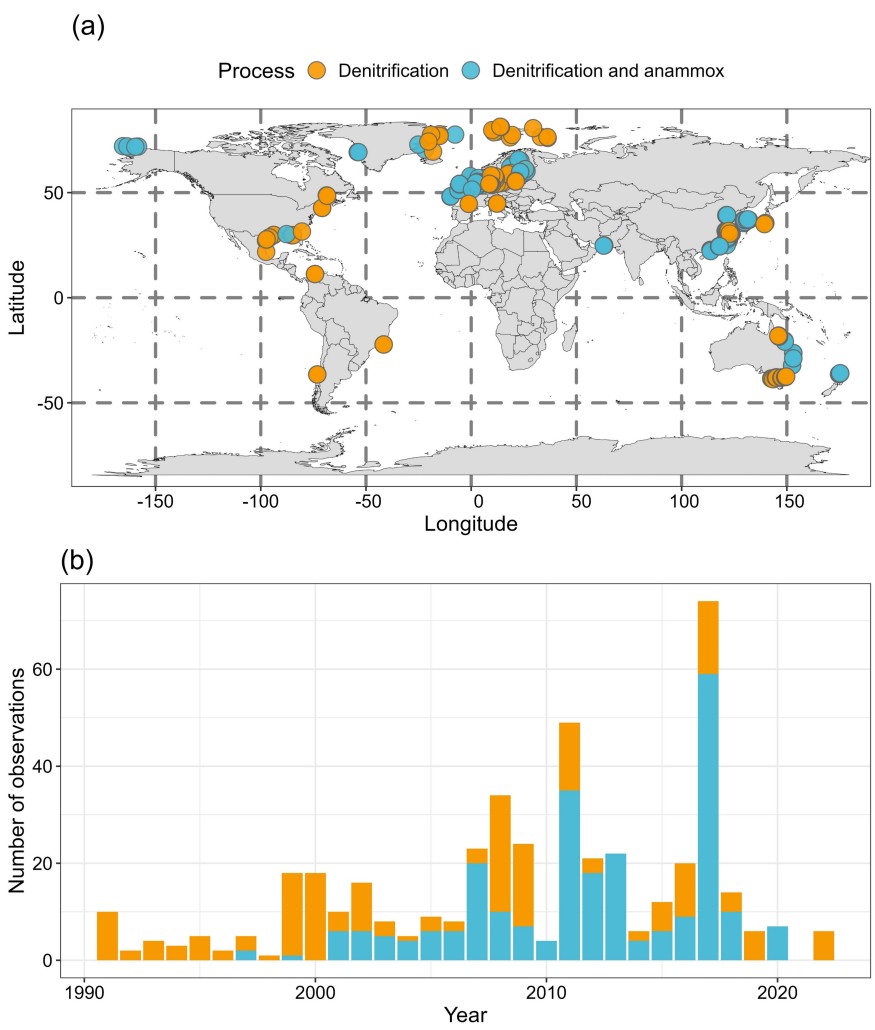

**Figure 1** Map showing the sampling sites distribution of nitrogen loss rate measurements (a) and the number of rate observations each year (b). Orange solid points denote that only denitrification rates were measured. Cyan solid points denote that both denitrification and anammox rates were measured.



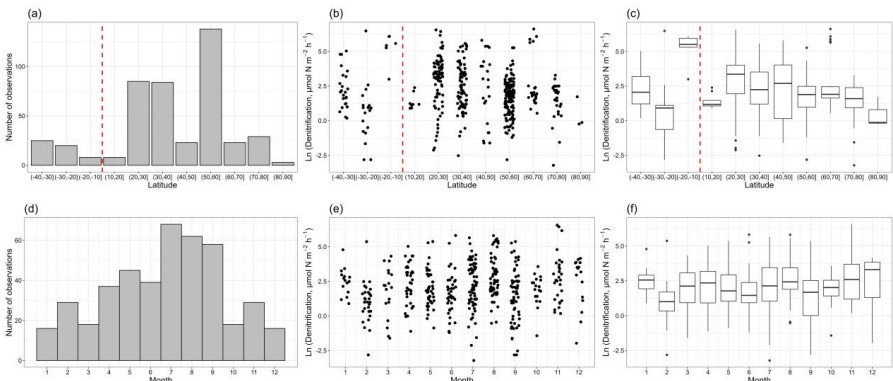

**Figure 2** The observation numbers of denitrification (a, d) and denitrification rates (b, c, e, f) with latitudinal bands and months. A vertical dashed red line delimits the Southern Hemisphere and the Northern Hemisphere. Tops and bottoms of boxes in box plots denote the 25th and 75th percentiles, respectively. The horizontal lines inside the box plots represent the medians. Whiskers mark the minimum and maximum values within 1.5 times the interquartile range, with black points representing outliers beyond that range.

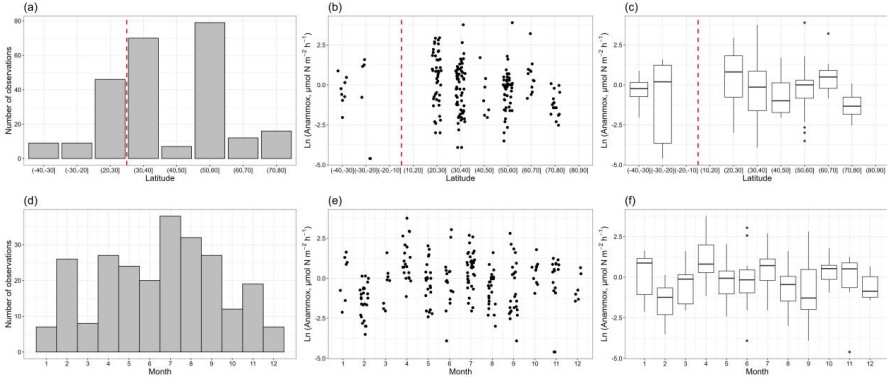

**Figure 3** The observation numbers of anammox (a, d) and anammox rates (b, c, e, f) with latitudinal bands and months.





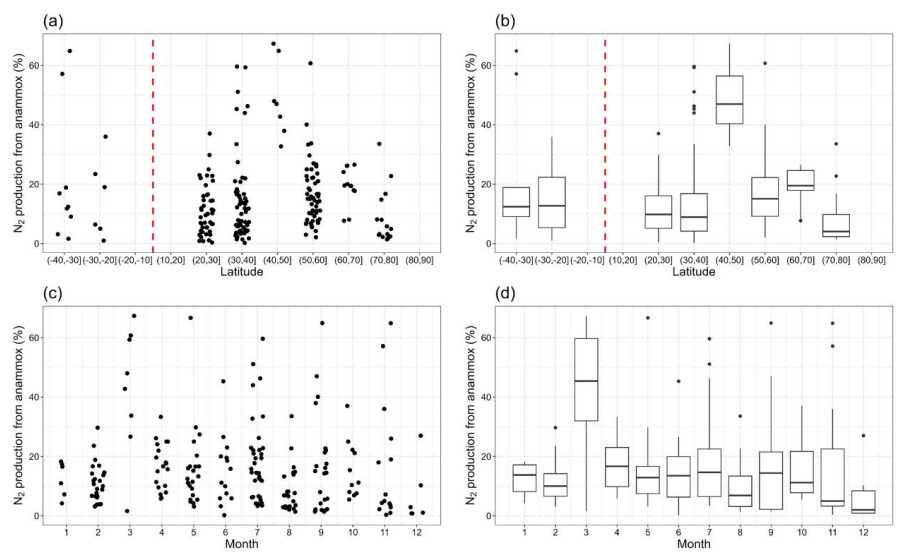

**Figure 4** The contribution of anammox to total N$_2$ production with latitudinal bands (a, b) and months (c, d).

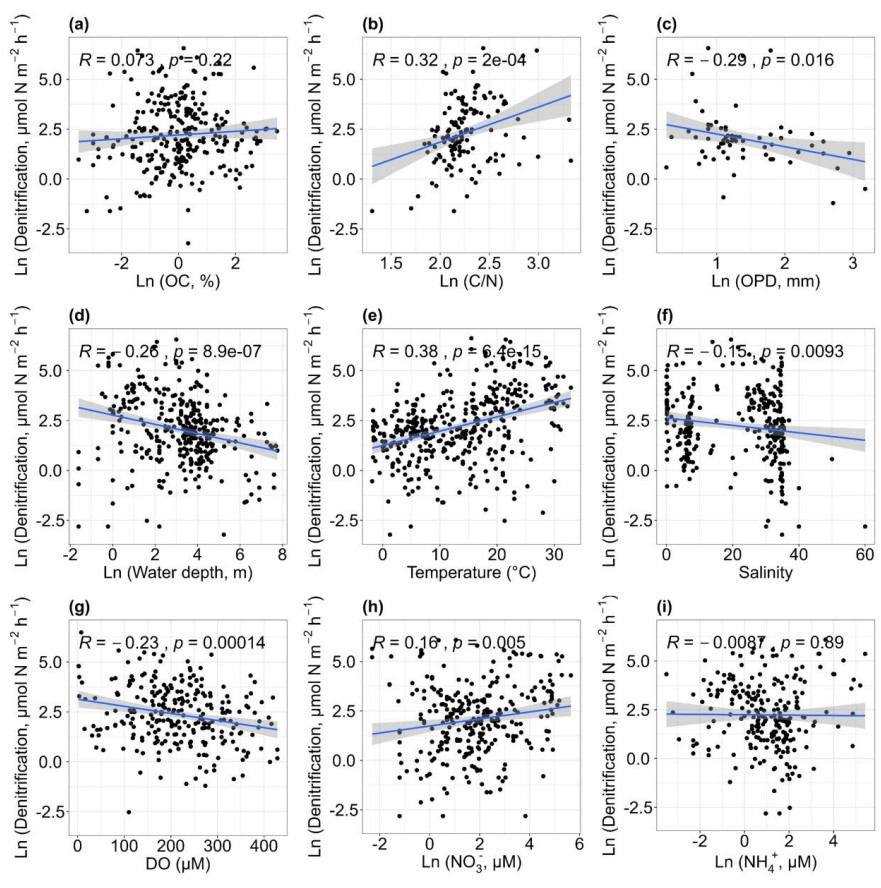

817

**Figure 5** Relationships between denitrification rates and organic carbon [OC, (a)], carbon-nitrogen ratios [C/N, (b)], oxygen penetration depth [OPD, (c)], water depth (d), temperature (e), salinity (f), dissolved oxygen [DO, (g)], nitrate concentrations [$NO_3^-$, (h)] and ammonium concentrations [$NH_4^+$, (i)].

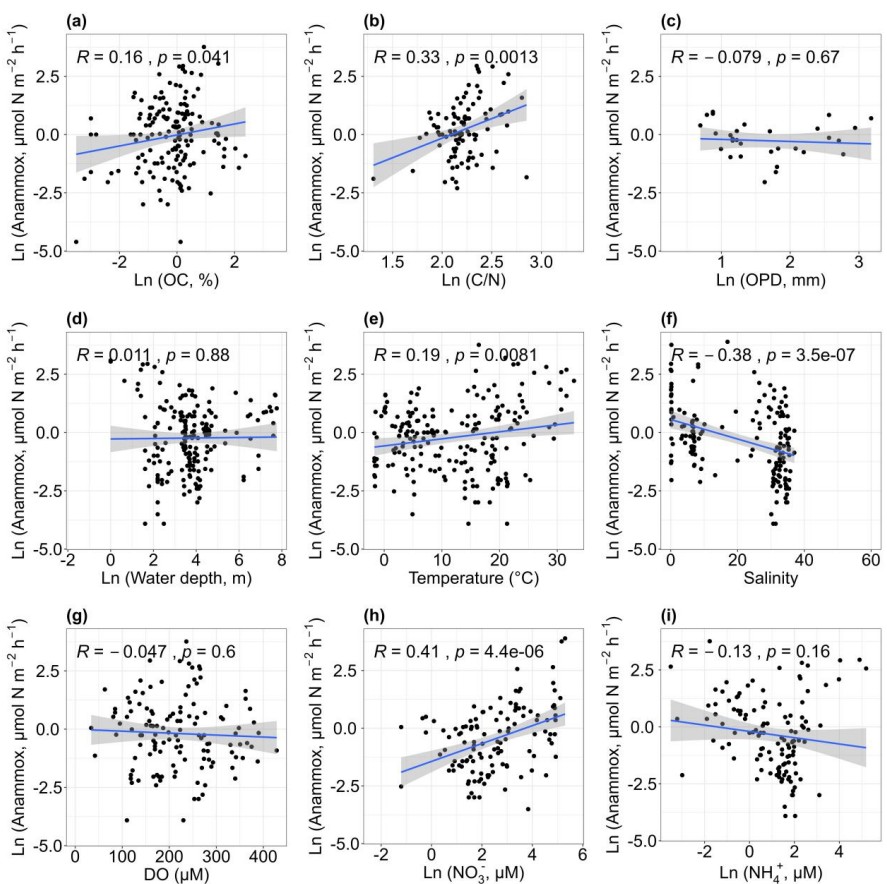

823

**Figure 6** Relationships between anammox rates and organic carbon [OC, (a)], carbon-nitrogen ratios [C/N, (b)], oxygen penetration depth [OPD, (c)], water depth (d), temperature (e), salinity (f), dissolved oxygen [DO, (g)], nitrate concentrations [$NO_3^-$, (h)] and ammonium concentrations [$NH_4^+$, (i)].

828



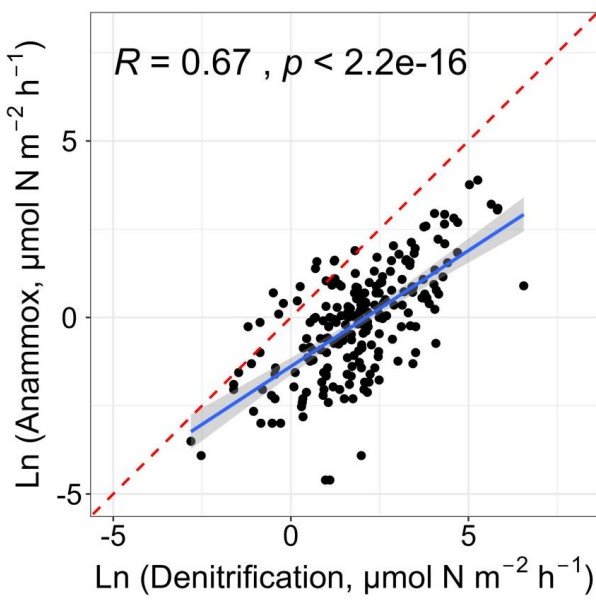

829

**Figure 7** Relationships between denitrification and anammox rates. The blue solid

line and red dashed line denote the linear regression and 1:1 line, respectively.

832

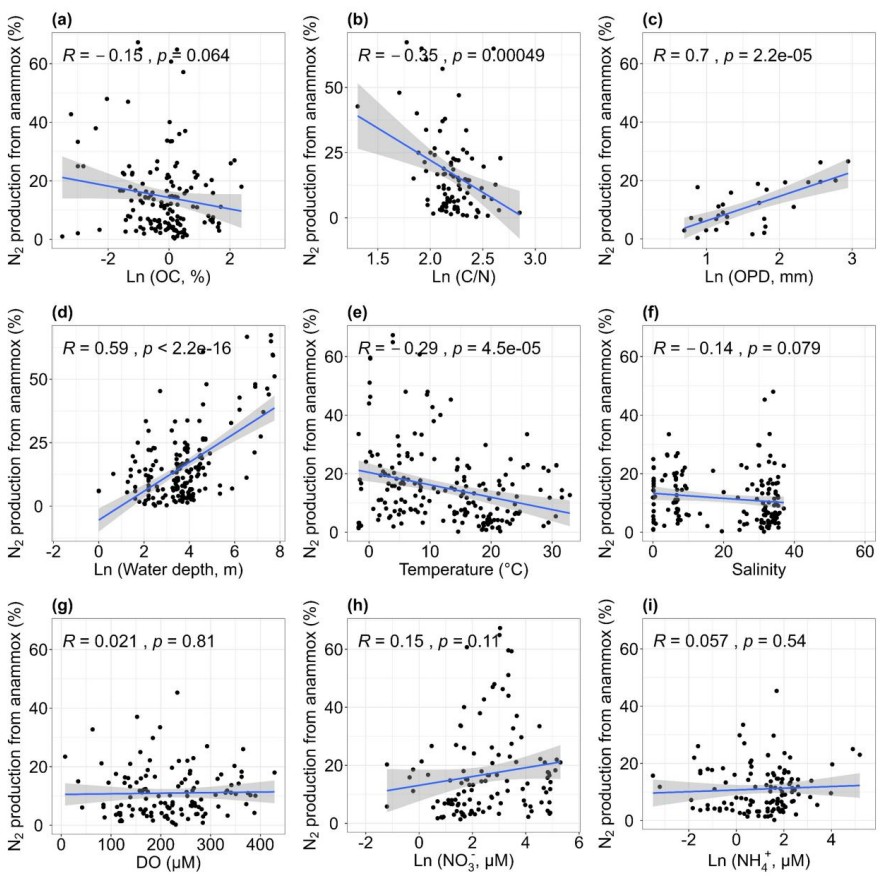

**Figure 8** Relationships between the relative contribution of anammox to total $N_2$ production and organic carbon [OC, (a)], carbon-nitrogen ratios [C/N, (b)], oxygen penetration depth [OPD, (c)], water depth (d), temperature (e), salinity (f), dissolved oxygen [DO, (g)], nitrate concentrations [$NO_3^-$, (h)] and ammonium concentrations [$NH_4^+$, (i)].



**Table 1** Summary of the observations of actual nitrogen loss rates. The locations,
observation numbers, core incubation methods and references are listed.

| Sampling locations | Observation numbers | Core incubations | References |
|---|---|---|---|
| Aarhus Bright, Denmark | 2 | Intact core incubations | (Nielsen and Glud, 1996) |
| Arabian Sea | 4 | Intact core incubations | (Sokoll et al., 2012) |
| Arctic fjord (Svalbard, Norway) | 3 | Intact core incubations | (Gihring et al., 2010b) |
| Bassin d'Arcachon coastal lagoon | 3 | Intact core incubations | (Welsh et al., 2000) |
| Casino, NSW, Australia | 2 | Intact core incubations | (Erler et al., 2008) |
| central Sagami Bay, Japan | 1 | Intact core incubations | (Glud et al., 2009) |
| Changjiang estuary and its adjacent East China Sea | 7 | Intact core incubations | (Song et al., 2021) |
| Changjiang River Estuary and Jiulong River Estuary, China | 23 | Intact core incubations | (Tan et al., 2022) |
| Changjiang River Estuary, China | 22 | Continuous-flow experiments | (Liu et al., 2020) |
| Changjiang River Estuary, China | 14 | Continuous-flow experiments | (Liu et al., 2019) |
| Coast of Finland, northern Baltic Sea | 10 | Intact core incubations | (Hellemann et al., 2020) |
| Coast of Victoria, Australia | 11 | Intact core incubations | (Kessler et al., 2018) |
| Coastal area of the Gulf of Gdańsk | 6 | Intact core incubations | (Benelli et al., 2024) |
| Coastal lagoons, France | 6 | Intact core incubations | (Rysgaard et al., 1996b) |
| Coastal sediments, Greenland | 11 | Intact core incubations | (Rysgaard et al., 2004) |
| Continental shelf and slope, North Atlantic | 12 | Intact core incubations | (Trimmer and Nicholls, 2009) |
| Continental shelf region off central Chile | 5 | Intact core incubations | (Farías et al., 2004) |
| Danshuei River in northern Taiwan, China | 1 | Intact core incubations | (Hsu and Kao, 2013) |
| East China Sea | 2 | Intact core incubations | (Song et al., 2016) |
| Elbe Estuary, North Frisian | 5 | Intact core | (Deek et al., |



| Location | Count | Method | Reference |
|---|---|---|---|
| Wadden Sea | | incubations | 2013) |
| Fjords in Svalbard and northern Norway | 5 | Intact core incubations | (Glud et al., 1998) |
| Georgia continental shelf, USA | 2 | Intact core incubations | (Vance-Harris and Ingall, 2005) |
| Great Barrier Reef lagoon | 2 | Intact core incubations | (Erler et al., 2013) |
| Gulf of Bothnia, Baltic Sea | 7 | Intact core incubations | (Bonaglia et al., 2017) |
| Gulf of Finland | 5 | Intact core incubations | (Susanna, 2007) |
| Gulf of Finland, Baltic Sea | 11 | Intact core incubations | (Jäntti and Hietanen, 2012) |
| Gulf of Finland, Baltic Sea | 13 | Intact core incubations | (Jäntti et al., 2011) |
| Gulf of Finland, Baltic Sea | 5 | Intact core incubations | (Hietanen and Kuparinen, 2008) |
| Gulf of Mexico | 6 | Intact core incubations | (Gihring et al., 2010a) |
| Gullmarsfjorden, Sweden and Thames Estuary, England | 2 | Intact core incubations | (Trimmer et al., 2006) |
| Hypoxic zone off the Changjiang River estuary, China | 9 | Intact core incubations | (Yang et al., 2022) |
| Jinpu Bay, China | 12 | Continuous-flow experiments | (Yin et al., 2015) |
| Jiulong River Estuary, China | 2 | Intact core incubations | (Wan et al., 2023) |
| Kattegat and Skagerrak | 10 | Intact core incubations | (Rysgaard et al., 2001) |
| Lawrence estuary | 1 | Intact core incubations | (Crowe et al., 2012) |
| Little Lagoon, USA | 1 | Continuous-flow experiments | (Bernard et al., 2015) |
| Noosa River estuary, Australia | 5 | Intact core incubations | (Chen et al., 2021) |
| North Sea | 9 | Intact core incubations | (Rosales Villa et al., 2019) |
| North Sea | 1 | Intact core incubations | (Fan et al., 2015) |
| North Sea | 8 | Intact core incubations | (Bale et al., 2014) |
| North Sea | 16 | Intact core incubations | (Neubacher et al., 2011) |
| Northeast Chukchi Sea | 5 | Continuous-flow | (McTigue et al., |



| | | | |
|---|---|---|---|
| | | | experiments | 2016) |
| Northeastern New Zealand continental shelf | 7 | Intact core incubations | (Cheung et al., 2024) |
| Northern Baltic Proper | 17 | Intact core incubations | (Bonaglia et al., 2014a) |
| Northern East China Sea, China | 16 | Continuous-flow experiments | (Chang et al., 2021) |
| Norwegian Trench, Skagerrak | 4 | Intact core incubations | (Trimmer et al., 2013) |
| Öre Estuary, Swedish | 6 | Intact core incubations | (Hellemann et al., 2017) |
| Pearl River Estuary, China | 5 | Intact core incubations | (Tan et al., 2019) |
| Plum Island Sound, Massachusetts | 4 | Intact core incubations | (Koop-Jakobsen and Giblin, 2010) |
| Randers Fjord and Norsminde Fjord, Denmark | 2 | Intact core incubations | (Risgaard-Petersen et al., 2004) |
| Randers Fjord, Young Sound and Skagerrak, Danmark | 3 | Intact core incubations | (Risgaard-Petersen et al., 2003) |
| Sacca di Goro lagoon, Italy | 6 | Intact core incubations | (Magri et al., 2020) |
| Southern and central Baltic Sea | 12 | Intact core incubations | (Deutsch et al., 2010) |
| Southern Finland | 5 | Intact core incubations | (Uusheimo et al., 2018) |
| St. George Island, Gulf of Mexico, Hausstrand, German Wadden Sea and Spitsbergen island, Svalbard | 5 | Intact core incubations | (Canion et al., 2014) |
| St. Joseph Bay, USA | 4 | Continuous-flow experiments | (Hoffman et al., 2019) |
| St. Lawrence Estuary, Canada | 3 | Intact core incubations | (Poulin et al., 2007) |
| Stockholm Archipelago, Baltic Sea | 1 | Intact core incubations | (Bonaglia et al., 2014b) |
| Svalbard, Norway | 10 | Intact core incubations | (Blackburn et al., 1996) |
| Taganga Bay, Colombia Caribbean | 8 | Intact core incubations | (Arroyave Gómez et al., 2020) |
| Tama Estuary, Japan | 2 | Continuous-flow experiments | (Usui et al., 2001) |
| Texas estuaries, USA | 26 | Continuous-flow experiments | (Gardner et al., 2006) |
| The Baltic Sea | 1 | Intact core | (Bonaglia et al., |



| | | incubations | 2013) |
|---|---|---|---|
| The Curonian Lagoon | 8 | Intact core incubations | (Bartoli et al., 2021) |
| Tropical Coastal Lagoons | 11 | Intact core incubations | (Enrich-Prast et al., 2016a) |
| Tropical Coastal Wetlands, Australia | 8 | Intact core incubations | (Adame et al., 2019b) |
| Ulleung Basin, East Sea | 9 | Intact core incubations | (Na et al., 2018) |
| Wallis Lake estuary, Australia | 2 | Intact core incubations | (Erler et al., 2017) |
| Young Sound fjord, northeast Greenland | 1 | Intact core incubations | (Rysgaard et al., 1996a) |

Continuous-flow experiments denote continuous flow experiments combined with core
incubations