# Peer review of "Global database of actual nitrogen loss rates in coastal and marine sediments"

_Earth System Science Data, 2024_

## Referee Comment (RC1)

**Review of Chang et al., ESSD submission**

The submission by Chang et al. presents an up-to-date compilation of nitrogen loss rates (denitrification and anammox) across various coastal systems. This topic is particularly important for understanding the biogeochemical cycles in marine environments, especially in the context of increasing anthropogenic nitrogen inputs. The authors had an extensive literature review and employed rigorous quality control measures to ensure the reliability of the database. The manuscript highlights the spatial and temporal distribution of denitrification and anammox, as well as the factors that influence these processes. The authors briefly introduced the isotope paring technique (IPT) to quantify nitrogen loss rates, providing a robust methodological framework for future research. The database offers a valuable resource for the scientific community.

As a dataset paper, careful consideration must be given to the potential biases in the data, such as the overrepresentation of certain regions and the exclusion of studies that did not report environmental variables. Below, I provide my comments and suggestions to further improve the manuscript:

Firstly, the dataset can be expanded, particularly for those measured via slurry incubation. At present, only whole core incubation data is included, which may not be sufficient to fully capture the general phenomena in anammox and denitrification experiments. Slurry incubation can be useful especially in teasing combined environmental effects. And I don't think the authors are making a good argument to exclude slurry incubation data (line 69). This limitation may result in specific findings with little global significance, such as the increase in the proportion of anammox in March and the higher denitrification rates in sediments with high carbon-to-nitrogen (C/N) ratios. The authors should better examine the dataset to minimize the bias from specific study sites.

Secondly, the authors should conduct a thorough examination of the data and perform a more detailed analysis of sediment characteristics before undertaking correlation analyses. Some parameters may not be suitable for correlation analysis due to their complex interactions and potential confounding factors. For instance, the variation in the proportion of anammox may not be closely related to latitude, as suggested in the manuscript, but may instead be more closely associated with the physical and chemical properties of the sediments.

As this manuscript is about using coastal nitrogen loss datasets to infer environmental controls, I would hope the authors share their thoughts about linking existing modelling work to their dataset. Are observations consistent with model interpretations? How can future observations be better conducted?

Below are some minor issues:

Line 90: In fact, over the past decades, the modeling community has been working on quantifying the effects of environmental factors on sedimentary denitrification:

Middelburg et al., 1996
https://agupubs.onlinelibrary.wiley.com/doi/pdf/10.1029/96GB02562

Bohlen et al., 2012

https://agupubs.onlinelibrary.wiley.com/doi/full/10.1029/2011GB004198

Li et al., 2024

https://bg.copernicus.org/articles/21/4361/2024/

The authors could provide a statement describing what's known and unknown to the community.

Line 125 – 127: Some coastal zones are inhabited by plants and animals; in some cases, whole core incubation would exclude the effect of benthic fauna or bioturbation, and the nutrient and oxygen availabilities in the core might not reflect in situ. It would be better to have an explanation about excluding these studies.

Line 134- 135: What "unit conversion techniques" were performed? Please explain.

Line 336 – 338: The authors could provide detailed explanation about investigating organic matter quantity and quality affecting sedimentary annamox. The current dataset is not supporting the idea.

Line 357 – 390: The discussion of the relationship between denitrification and anammox rates could be more concise and focused.

Section 4: This section is important and can be improved. Readers may want to know more about the potential applications of this database, and specific examples of how the data can be used in future studies.

Section 5: The conclusion could be more forward-looking, emphasizing the potential for future research and applications.

Figure 2: Figure titles are too complex, please enhance the clarity of the figure window labels and descriptions. The box plots show the median, interquartile range, and outliers for each latitudinal band and month.

---

## Referee Comment (RC2)

Reviewer's comments for "Global database of actual nitrogen loss rates in coastal and marine sediments".

This study compiles global denitrification and anammox data from both open ocean and estuarine environments, providing a valuable dataset for the scientific community, particularly for researchers studying the nitrogen cycle. The database offers insights into nitrogen loss processes and their environmental controls, which can support future studies and biogeochemical modelling. While the study is well-organized, some aspects require clarification. Below are my comments and suggestions

Line 55 The full name of anammox (Anaerobic Ammonium Oxidation) should be provided here.

Line 69 Please provide a brief introduction to slurry incubation and intact core incubation methods to clarify their differences and applications.

Line 109 Please provide a brief introduction to continuous flow experiments to clarify their methodology.

Line 113 Slurry incubation provides valuable data in certain aspects, and completely excluding these measurements may not be appropriate.

Line 119 Please clarify why measurements under light incubation were excluded.

Line 170 Consider adding a figure to summarize the calculation methods for better clarity

Line 286 Other factors, such as iron and sulfide, can also influence denitrification and anammox. Why were these not considered? While some studies may not have measured these parameters, it would be valuable to discuss their potential role.

In addition, this section applies multiple regression analyses to explore the relationships between various controlling factors and denitrification/anammox. I am curious whether the authors were able to determine a threshold value for these factors—beyond which denitrification exceeds anammox. Additionally, based on the compiled data, which parameter is identified as the most significant controlling factor

Line 357 I am wondering about the sediment characteristics at these study sites. Do they include vegetated areas? These factors can significantly influence denitrification and anammox rates.

Some data in the table represent open ocean environments, while others are from riverine systems. Please consider adding water depth to Table 1 to provide clearer context for the different study sites.

---

## Author Comment (AC1)

Response to essd-2024-539 RC1:

We first thank the Reviewer 1 for the thorough review of this manuscript. The feedback provided constructive comments and suggestions and incorporating the feedback to this draft will improve the quality of this work greatly.

Review of Chang et al., ESSD submission

The submission by Chang et al. presents an up-to-date compilation of nitrogen loss rates (denitrification and anammox) across various coastal systems. This topic is particularly important for understanding the biogeochemical cycles in marine environments, especially in the context of increasing anthropogenic nitrogen inputs. The authors had an extensive literature review and employed rigorous quality control measures to ensure the reliability of the database. The manuscript highlights the spatial and temporal distribution of denitrification and anammox, as well as the factors that influence these processes. The authors briefly introduced the isotope paring technique (IPT) to quantify nitrogen loss rates, providing a robust methodological framework for future research. The database offers a valuable resource for the scientific community.

Thank you for your positive comments.

As a dataset paper, careful consideration must be given to the potential biases in the data, such as the overrepresentation of certain regions and the exclusion of studies that did not report environmental variables. Below, I provide my comments and suggestions to further improve the manuscript:

Response: Thanks for this comment. We have checked the database and answered the question on the overrepresentation of certain regions in the following point-to-point response. We do exclude those studies that did not report environmental variables. In line 128, we addressed this point "At least one environmental variable was recorded for each selected study..."

Firstly, the dataset can be expanded, particularly for those measured via slurry incubation. At present, only whole core incubation data is included, which may not be sufficient to fully capture the general phenomena in anammox and denitrification experiments. Slurry incubation can be useful especially in teasing combined environmental effects. And I don't think the authors are making a good argument to exclude slurry incubation data (line 69). This limitation may result in specific findings with little global significance, such as the increase in the proportion of anammox in March and the higher denitrification rates in sediments with high carbon-to-nitrogen (C/N) ratios. The authors should better examine the dataset to minimize the bias from specific study sites.

Response: Thank you for the suggestions and insightful comments. We know slurry incubation experiments have advantages in teasing key environmental factors (Brin et al., 2017; Deng et al., 2015; Tan et al., 2020). We once considered integrating slurry incubation data and whole core incubation data, however, considering the different meanings of rates by slurry incubation and whole core incubation experiments and the differences in potential and actual rates calculation, we did not include slurry incubation data. Firstly, the rates obtained from slurry incubation represent the potential rate, and the

whole core incubation can obtain the actual rates. Secondly, some potential rate calculations include both $^{14}$N-based and $^{15}$N-based rates (Na et al., 2018; Thamdrup and Dalsgaard, 2002), while others only include $^{15}$N-based rates (Deng et al., 2015), making it difficult to make comparisons.

Brin, L.D., Giblin, A.E., Rich, J.J. 2017. Similar temperature responses suggest future climate warming will not alter partitioning between denitrification and anammox in temperate marine sediments. Global Change Biology, 23(1), 331-340.

Deng, F., Hou, L., Liu, M., Zheng, Y., Yin, G., Li, X., Lin, X., Chen, F., Gao, J., Jiang, X. 2015. Dissimilatory nitrate reduction processes and associated contribution to nitrogen removal in sediments of the Yangtze Estuary. Journal of Geophysical Research: Biogeosciences, 120(8), 1521-1531.

Na, T., Thamdrup, B., Kim, B., Kim, S.-H., Vandieken, V., Kang, D.-J. and Hyun, J.-H. 2018. N$_2$ production through denitrification and anammox across the continental margin (shelf–slope–rise) of the Ulleung Basin, East Sea. Limnology and Oceanography, 63, S410-S424.

Tan, E., Zou, W., Zheng, Z., Yan, X., Du, M., Hsu, T.-C., Tian, L., Middelburg, J.J., Trull, T.W., Kao, S.-j. 2020. Warming stimulates sediment denitrification at the expense of anaerobic ammonium oxidation. Nature Climate Change, 10(4), 349-355.

Thamdrup, B., Dalsgaard, T. 2002. Production of N$_2$ through anaerobic ammonium oxidation coupled to nitrate reduction in marine sediments. Applied and Environmental Microbiology, 68, 1312–1318.

Additionally, a recent study has already summarized the data on nitrogen loss rates by slurry incubations in aquatic systems. We have added this sentence in line 121-126. "Given a recent study has already summarized the data on nitrogen loss rates by slurry incubations in aquatic systems (He et al., 2025), this work only selected data in which denitrification and/or anammox rates were measured using intact core incubations with 15N isotope pairing techniques, excluding measurements derived from slurry incubations."

He, G., Deng, D., Delgado-Baquerizo, M., Liu, W., and Zhang, Q.: Global Relative Importance of Denitrification and Anammox in Microbial Nitrogen Loss Across Terrestrial and Aquatic Ecosystems, Advanced Science, 12, 2406857, https://doi.org/10.1002/advs.202406857, 2025.

To exclude slurry incubation data (line 69), we modified the following expression (line 72-78). "Slurry incubations have been used to estimate the potential rates and have advantages in discovering nitrogen loss processes in the environment (Thamdrup & Dalsgaard, 2002) as well as studying the environmental controls of these pathways, however, the natural gradients of substrates and redox in sediments were disrupted during incubations (Trimmer et al., 2006). The intact core incubations can quantify nitrogen removal processes in intact sediments and reflect the genuine benthic nitrogen transformation rates. The application of intact core incubations will enable us..."

Thank you for this suggestion to examine the dataset, we have checked the whole dataset

to address the following questions.

For the increase in the proportion of anammox in March, we noted these observations were mainly distributed in certain regions, so the extrapolation of this result should be cautious. We have amended the following sentences in line 311-314. "It is worth noting that the rate observations in March were mainly distributed in certain regions. Thus, the extrapolations of relative importance of anammox in coastal marine ecosystems at the monthly level using this result should be cautious. More observation data in other regions are needed in the future."

For the higher denitrification rates in sediments with high carbon-to-nitrogen (C/N) ratios, we have checked the relevant data and found these observations were not limited to specific study sites. Furthermore, even after removing high rate values, this correlation still exists. The possible reason was given in line 333-335. "One possible explanation is that microbial communities may adapt to use organic matter typically encountered, though the organic matter is not labile (Salk et al., 2017)."

Secondly, the authors should conduct a thorough examination of the data and perform a more detailed analysis of sediment characteristics before undertaking correlation analyses. Some parameters may not be suitable for correlation analysis due to their complex interactions and potential confounding factors. For instance, the variation in the proportion of anammox may not be closely related to latitude, as suggested in the manuscript, but may instead be more closely associated with the physical and chemical properties of the sediments.

Response: Thank you for this suggestion. We conducted a thorough examination of the data prior to performing correlation analyses. Firstly, we identified missing values across multiple environmental variables. Secondly, during correlation analyses involving these variables, missing data were systematically excluded. Following these steps, only a limited number of observations remained available for environmental variables. Due to the reduced sample size and potential confounding factors, we refrained from conducting correlation analyses between environmental variables themselves. The proportion of anammox changed along the latitude, reflecting the spatial distribution in proportion of anammox, and may be regulated by physical and chemical properties of the sediments, indicating the environmental controls. In order to better present the data set, we made simple correlation analysis to show the relations between nitrogen loss rates and environmental variables, and didn't exclude parameters.

As this manuscript is about using coastal nitrogen loss datasets to infer environmental controls, I would hope the authors share their thoughts about linking existing modelling work to their dataset. Are observations consistent with model interpretations? How can future observations be better conducted?

Response: Thank you for this suggestion. Here we focus on describing the existing integrated data and conducting a simple linear analysis between nitrogen loss data and environmental factors. For some environmental variables, the observations are consistent with model interpretations. For example, Middelburg et al. (1996) have pointed out that the nitrate content and dissolved oxygen of the bottom water are important regulatory factors,

our results are in agreement with their findings. The functions of model are based on empirical data, and observations provide data support for developing new empirical formulas of model. This dataset can provide new parameters for future predictions, and suitable parameters from observations can be integrated into the existing model to provide better constraints on nitrogen loss processes in coastal marine systems in the future.

For future observations, in terms of spatial scale, more studies are needed in areas with limited observation data on nitrogen loss rates, and in terms of temporal scale, there is an urgent need to conduct more research in month with limited data to deepen our understanding in the nitrogen cycle. Additionally, when studying nitrogen loss rates, particular attention should be paid to enhancing the monitoring of multiple environmental parameters. See the relevant contents added in Section 4 in line 482-486.

Middelburg, J. J., Soetaert, K., Herman, P. M. J., and Heip, C. H. R.: Denitrification in marine sediments: A model study, Global Biogeochem. Cycles, 10, 661-673, https://doi.org/10.1029/96GB02562, 1996.

Below are some minor issues:

Line 90: In fact, over the past decades, the modeling community has been working on quantifying the effects of environmental factors on sedimentary denitrification:
Middelburg et al., 1996
https://agupubs.onlinelibrary.wiley.com/doi/pdf/10.1029/96GB02562
Bohlen et al., 2012
https://agupubs.onlinelibrary.wiley.com/doi/full/10.1029/2011GB004198
Li et al., 2024
https://bg.copernicus.org/articles/21/4361/2024/
The authors could provide a statement describing what's known and unknown to the community.

Response: Thank you for this suggestion. We have amended the following sentence in line 91-94. "The modeling community also has conducted many researches on environmental regulation of nitrogen loss (mainly denitrification), and improved the predictive parameters of denitrification (Middelburg et al., 1996; Bohlen et al., 2012; Li et al., 2024)."

Bohlen, L., Dale, A. W., and Wallmann, K.: Simple transfer functions for calculating benthic fixed nitrogen losses and C:N:P regeneration ratios in global biogeochemical models, Global Biogeochem. Cycles, 26, https://doi.org/10.1029/2011GB004198, 2012.

Li, N., Somes, C. J., Landolfi, A., Chien, C. T., Pahlow, M., and Oschlies, A.: Global impact of benthic denitrification on marine $N_2$ fixation and primary production simulated by a variable-stoichiometry Earth system model, Biogeosciences, 21, 4361-4380, https://doi.org/10.5194/bg-21-4361-2024, 2024.

Middelburg, J. J., Soetaert, K., Herman, P. M. J., and Heip, C. H. R.: Denitrification in marine sediments: A model study, Global Biogeochem. Cycles, 10, 661-673,

https://doi.org/10.1029/96GB02562, 1996.

Line 125 – 127: Some coastal zones are inhabited by plants and animals; in some cases, whole core incubation would exclude the effect of benthic fauna or bioturbation, and the nutrient and oxygen availabilities in the core might not reflect in situ. It would be better to have an explanation about excluding these studies.

Response: Thank you for this suggestion. We have made some explanations and added these sentences in line138-144. "Some coastal zones are inhabited by plants and animals, whole core incubation would exclude the effect of benthic fauna or bioturbation as the nutrient and oxygen availabilities in the core might not reflect *in situ* sediment characteristics. In addition, whole core incubation would exclude the effect of antibiotics addition because antibiotics addition could influence *in situ* nitrogen removal rates (Wan et al., 2023). Thus, studies examining the effects of meiofauna or antibiotics on nitrogen removal were not included ..."

Line 134- 135: What "unit conversion techniques" were performed? Please explain.

Response: Thank you for this suggestion. We have made supplements after this sentence in line 153-155. "For example, nitrogen loss (including denitrification and anammox) rates were in $\mu$mol N m$^{-2}$ h$^{-1}$. When rates in the texts were displayed as mmol N m$^{-2}$ d$^{-1}$ or $\mu$mol N m$^{-2}$ d$^{-1}$, they were converted to $\mu$mol N m$^{-2}$ h$^{-1}$."

Line 336 – 338: The authors could provide detailed explanation about investigating organic matter quantity and quality affecting sedimentary annamox. The current dataset is not supporting the idea.

Response: Thank you for this suggestion. We have revised this sentence and made some explanations in line 365-370. "We infer that, to some extent, the coupling of denitrification and anammox may account for this relation. As mentioned above, denitrification stimulated with higher C/N ratios, decomposition of organic matter could provide substrate for anammox, thereby promoting anammox. More research is needed to reveal the influencing mechanisms of C/N ratios on anammox."

Line 357 – 390: The discussion of the relationship between denitrification and anammox rates could be more concise and focused.

Response: Thank you for this suggestion. Line 357 – 390, here two parts of discussion were reported. Line 357 – 373, this paragraph discussed the relationship between denitrification and anammox rates. Line 373 – 390, this paragraph discussed the relationship between the contribution of anammox to total N$_2$ production (ra) and physicochemical parameters. We have compressed the discussion of this section.

Section 4: This section is important and can be improved. Readers may want to know more about the potential applications of this database, and specific examples of how the data can be used in future studies.

Response: Thanks for this suggestion. We have made some additions. In application (2) part, "Note that environmental variables have missing values, which limits our analysis of

environmental factors affecting nitrogen loss rates. For better studying the environmental controlls, these missing values can be filled using the multivariate imputation with random forests method (Hou et al., 2021)." In application (3) part, "For example, by integrating potential key factors of nitrogen removal processes into machine learning architectures, future studies can develop spatially predictive models for global nitrogen loss rates by the references of Laffitte et al. (2025) and Ling et al. (2025)." In application (4) part, "The previous model considered constraint parameters such as nitrate, dissolved oxygen, chlorophyll, and phosphate content (Middelburg et al., 1996; Bohlen et al., 2012; Li et al., 2024), and other parameters provided in this dataset can supply new parameter supplements for the biogeochemical model." In application (5) part, "(5) guiding future observations. More studies are needed in areas and months with limited observation data on nitrogen loss rates to deepen our understanding of the nitrogen cycle worldwide. Additionally, when studying nitrogen loss rates, particular attention should be paid to enhancing the monitoring of multiple environmental parameters."

Hou, E., Wen, D., Jiang, L., Luo, X., Kuang, Y., Lu, X., Chen, C., Allen, K. T., He, X., Huang, X., and Luo, Y.: Latitudinal patterns of terrestrial phosphorus limitation over the globe, Ecology Letters, 24, 1420-1431, https://doi.org/10.1111/ele.13761, 2021.

Laffitte, B., Zhou, T., Yang, Z., Ciais, P., Jian, J., Huang, N., Seyler, B. C., Pei, X., and Tang, X.: Timescale Matters: Finer Temporal Resolution Influences Driver Contributions to Global Soil Respiration, Global Change Biol., 31, e70118, https://doi.org/10.1111/gcb.70118, 2025.

Ling, J., Dungait, J. A. J., Delgado-Baquerizo, M., Cui, Z., Zhou, R., Zhang, W., Gao, Q., Chen, Y., Yue, S., Kuzyakov, Y., Zhang, F., Chen, X., and Tian, J.: Soil organic carbon thresholds control fertilizer effects on carbon accrual in croplands worldwide, Nat. Commun., 16, 3009, https://doi.org/10.1038/s41467-025-57981-6, 2025.

Section 5: The conclusion could be more forward-looking, emphasizing the potential for future research and applications.

Response: Thank you for this suggestion . We have emphasized the potential for future research and applications by adding the following sentences. " The establishment of this global database on denitrification and anammox in coastal and marine sediments provides a critical foundation for advancing nitrogen cycle research and generating novel insights. This database enables the comparison of these two nitrogen loss processes, evaluation of the environmental controls across spatial scales (local to global), prediction of the global biogeography of denitrification and anammox, parameterization and development of biogeochemical models, and guide direction of observations in the future."

Figure 2: Figure titles are too complex, please enhance the clarity of the figure window labels and descriptions. The box plots show the median, interquartile range, and outliers for each latitudinal band and month.

Response: Thank you for this suggestion. We have enhanced the clarity of the figure window labels and adopted your suggestion on the box plots description. "The box plots show the median, interquartile range, and outliers for each latitudinal band and month." We also have enhanced the clarity of the figure window labels in Figure 3 and Figure 4.

---

## Author Comment (AC2)

**Response to essd-2024-539 RC2:**

We first thank the Reviewer 2 for the thorough review of this manuscript. The feedback provided constructive comments and suggestions and incorporating the feedback to this draft will improve the quality of this work greatly.

Reviewer's comments for "Global database of actual nitrogen loss rates in coastal and marine sediments".

This study compiles global denitrification and anammox data from both open ocean and estuarine environments, providing a valuable dataset for the scientific community, particularly for researchers studying the nitrogen cycle. The database offers insights into nitrogen loss processes and their environmental controls, which can support future studies and biogeochemical modelling. While the study is well-organized, some aspects require clarification. Below are my comments and suggestions

Thanks for your positive comments.

Line 55 The full name of anammox (Anaerobic Ammonium Oxidation) should be provided here.

Response: Thank you for this suggestion. We have added it.

Line 69 Please provide a brief introduction to slurry incubation and intact core incubation methods to clarify their differences and applications.

Response: Thank you for this suggestion. We have improved the expression and provided a brief introduction to slurry incubation and intact core incubation in line 72-78. "Slurry incubations have been used to estimate the potential rates, and have advantages in discovering nitrogen loss processes in the environment (Thamdrup & Dalsgaard, 2002) as well as studying the environmental controls of nitrogen loss pathways, however, the natural gradients of substrates and redox in sediments were disrupted during incubations (Trimmer et al., 2006). The intact core incubations can quantify nitrogen removal processes in intact sediments and reflect the genuine benthic nitrogen transformation rates. The application of intact core incubations will enable us..."

Line 109 Please provide a brief introduction to continuous flow experiments to clarify their methodology.

Response: Thank you for this suggestion. We have amended a brief introduction in line 115-119. "For continuous flow experiments, incubations were carried out in a flow-through system where bottom water was pumped over intact cores using a multi-channel peristaltic pump, and inflow and outflow samples were collected to quantify the nitrogen process rates after the addition of 15N tracer (Gardner & McCarthy, 2009)."

Gardner, W. S. and McCarthy, M. J.: Nitrogen dynamics at the sediment–water interface in shallow, sub-tropical Florida Bay: why denitrification efficiency may decrease with increased eutrophication, Biogeochemistry, 95, 185-198, https://doi.org/10.1007/s10533-009-9329-5, 2009. Line 113 Slurry incubation provides valuable data in certain aspects, and completely excluding these measurements may not be appropriate.

Response: Thank you for this comment. We know that slurry incubation provides valuable data and has advantage in certain aspects. We once considered integrating slurry incubation data and whole core incubation data, however, considering the different meanings of rates by slurry incubation and whole core incubation experiments and the differences in potential and actual rates calculation, we did not include slurry incubation data. Firstly, the rates obtained from slurry incubation represent the potential rate, and the whole core incubation can obtain the actual rates. Secondly, some potential rate calculations include both 14N-based and 15N-based rates (Na et al., 2018; Thamdrup and Dalsgaard, 2002), while others only include 15N-based rates (Deng et al., 2015), making it difficult to make comparisons.

- Deng, F., Hou, L., Liu, M., Zheng, Y., Yin, G., Li, X., Lin, X., Chen, F., Gao, J., Jiang, X. 2015. Dissimilatory nitrate reduction processes and associated contribution to nitrogen removal in sediments of the Yangtze Estuary. Journal of Geophysical Research: Biogeosciences, 120(8), 1521-1531.
- Na, T., Thamdrup, B., Kim, B., Kim, S.-H., Vandieken, V., Kang, D.-J. and Hyun, J.-H. 2018. N2 production through denitrification and anammox across the continental margin (shelf–slope–rise) of the Ulleung Basin, East Sea. Limnology and Oceanography, 63, S410-S424.
- Thamdrup, B., Dalsgaard, T. 2002. Production of N2 through anaerobic ammonium oxidation coupled to nitrate reduction in marine sediments. Applied and Environmental Microbiology, 68, 1312–1318.

Furthermore, a recent study has summarized the spatial distribution and drivers of nitrogen loss rates by slurry incubations in aquatic systems. To compare the differences in nitrogen removal rates determined by the same method, here we focus on rates measured by intact core incubation and exclude slurry incubation data. We have added this sentence in line 121-126. "Given a recent study has already summarized the data on nitrogen loss rates by slurry incubations in aquatic systems (He et al., 2025), this work only selected data in which denitrification and/or anammox rates were measured using intact core incubations with 15N isotope pairing techniques, excluding measurements derived from slurry incubations."

He, G., Deng, D., Delgado-Baquerizo, M., Liu, W., and Zhang, Q.: Global Relative Importance of Denitrification and Anammox in Microbial Nitrogen Loss Across Terrestrial and Aquatic Ecosystems, Advanced Science, 12, 2406857, https://doi.org/10.1002/advs.202406857, 2025.

**Line 119 Please clarify why measurements under light incubation were excluded.**

Response: Thank you for this suggestion. We have clarified the excluding reason in line 127-130. "Photosynthetic  $O_2$  production can influence  $O_2$  penetration depth and thereby nitrate availability in sediments, interfering with denitrification rates in the nitrate reduction zone (Chen et al., 2021; Bartoli et al., 2021). In cases where nitrogen loss rates were measured under both light and dark conditions, only those measured in the dark were

included to avoid photosynthesis and facilitate comparison with other studies."

Line 170 Consider adding a figure to summarize the calculation methods for better clarity Response: Thank you for this suggestion. For the calculation methods mentioned in this part, as Salk et al. (2017) have already summarized the different calculation methods on nitrogen loss rates and presented relevant pictures clearly, here we don't draw pictures. We have added this reference in this part for better clarity and made the following description in line 243-247. "Regarding the aforementioned calculation methods, Salk et al. (2017) have systematically reviewed different methods for quantifying nitrogen loss rates and illustrated their differences with diagrams distinguishing different processes, providing valuable guidance for researchers interested in this field. Therefore, interested researchers can refer to their article."

Salk, K. R., Erler, D. V., Eyre, B. D., Carlson-Perret, N., and Ostrom, N. E.: Unexpectedly high degree of anammox and DNRA in seagrass sediments: Description and application of a revised isotope pairing technique, Geochim. Cosmochim. Acta, 211, 64-78, https://doi.org/10.1016/j.gca.2017.05.012, 2017.

Line 286 Other factors, such as iron and sulfide, can also influence denitrification and anammox. Why were these not considered? While some studies may not have measured these parameters, it would be valuable to discuss their potential role.

Response: Thank you for this suggestion. As the reviewer pointed out, due to the limited data on iron and sulfide, we did not include these data in the database. We have discussed their potential influence on denitrification and anammox in line 394-417.

"Other factors, such as iron, manganese, and sulfide, although not included in the database, can also influence denitrification and anammox rates. For example, Fe oxides were observed to be positively correlated with denitrification rates in the Jinpu Bay, China (Yin et al., 2015). The mechanism may be that ferrous iron can supply an electron donor for nitrate, thereby promoting denitrification. Anschutz et al. (2000) found manganese dioxides could also serve as electron donors for denitrification. Deng et al. (2015) showed a positive relationship between denitrification rates and sulfide concentrations in the Changjiang Estuary sediments, revealing that sulfide can act as energy sources for denitrification. In contrast, evidence has shown that sulfide exerts inhibitory effects on nitrogen removal in coastal sediments by inhibiting the metabolism of denitrifying microorganisms (Aelion and Warttinger, 2010). Thus, the impact of sulfide on denitrification remains controversial. For anammox, a study found that sulfide could affect anammox activity. Yin et al. (2015) found that anammox rates were positively correlated with sulfide concentrations. This phenomenon is likely attributed to sulfide-induced nitrite accumulation during incomplete denitrification processes, where sulfide inhibits the activity of nitric oxide reductase and nitrous oxide reductase, thereby enhancing anammox activity. Under anaerobic conditions, ammonium oxidation can be coupled with the reduction of ferric iron, sulfate, and Mn(IV)-oxides. For example, Rios-Del Toro et al. (2018) confirmed that ammonium oxidation was associated with ferric iron and sulfate reduction under anaerobic conditions, thereby stimulating nitrogen loss in marine sediments. Evidence shows ammonium loss is coupled with Fe(III) and Mn(IV) reduction in coastal environments (Samperio-Ramos et al., 2024), demonstrating the crucial roles of metal oxides in removing reactive nitrogen."

- Aelion, C. M. and Warttinger, U.: Sulfide Inhibition of Nitrate Removal in Coastal Sediments, Estuaries Coasts, 33, 798-803, https://doi.org/10.1007/s12237-010-9275-4, 2010.
- Anschutz, P., Sundby, B., Lefrançois, L., Luther, G. W., and Mucci, A.: Interactions between metal oxides and species of nitrogen and iodine in bioturbated marine sediments, Geochim. Cosmochim. Acta, 64, 2751-2763, https://doi.org/10.1016/S0016-7037(00)00400-2, 2000.
- Deng, F., Hou, L., Liu, M., Zheng, Y., Yin, G., Li, X., Lin, X., Chen, F., Gao, J., and Jiang, X.: Dissimilatory nitrate reduction processes and associated contribution to nitrogen removal in sediments of the Yangtze Estuary, J. Geophys. Res.:Biogeosci., 120, 1521-1531, https://doi.org/10.1002/2015JG003007, 2015.
- Rios-Del Toro, E. E., Valenzuela, E. I., López-Lozano, N. E., Cortés-Martínez, M. G., Sánchez-Rodríguez, M. A., Calvario-Martínez, O., Sánchez-Carrillo, S., and Cervantes, F. J.: Anaerobic ammonium oxidation linked to sulfate and ferric iron reduction fuels nitrogen loss in marine sediments, Biodegradation, 29, 429-442, https://doi.org/10.1007/s10532-018-9839-8, 2018.
- Samperio-Ramos, G., Hernández-Sánchez, O., Camacho-Ibar, V. F., Pajares, S., Gutiérrez, A., Sandoval-Gil, J. M., Reyes, M., De Gyves, S., Balint, S., Oczkowski, A., Ponce-Jahen, S. J., and Cervantes, F. J.: Ammonium loss microbiologically mediated by Fe(III) and Mn(IV) reduction along a coastal lagoon system, Chemosphere, 349, 140933, https://doi.org/10.1016/j.chemosphere.2023.140933, 2024.

In addition, this section applies multiple regression analyses to explore the relationships between various controlling factors and denitrification/anammox. I am curious whether the authors were able to determine a threshold value for these factors—beyond which denitrification exceeds anammox. Additionally, based on the compiled data, which parameter is identified as the most significant controlling factor

Response: Thank you for this suggestion. Here we use simple correlation analyses to explore the relationships instead of multiple regression analyses. In most cases denitrification exceeds anammox as we can see from Figure 4 (line 941). From the perspective of nitrogen loss percentage, denitrification generally dominates nitrogen loss processes. In contrast, anammox exceeds denitrification mainly in stations with water depths between 100 and 2342 m, including a continental shelf to slope transect in the North Atlantic (Trimmer and Nicholls, 2009), the deep Norwegian Trench in the Skagerrak (Trimmer et al., 2013), and the continental margin (shelf – slope – rise) of the Ulleung Basin, East Sea (Na et al., 2018). Thus, by far we have been unable to determine a threshold value for these factors—beyond which denitrification exceeds anammox.

Na, T., Thamdrup, B., Kim, B., Kim, S.-H., Vandieken, V., Kang, D.-J., and Hyun, J.-H.: N2 production through denitrification and anammox across the continental margin (shelf - slope - rise) of the Ulleung Basin, East Sea, Limnol. Oceanogr., 63, S410-S424, https://doi.org/10.1002/lno.10750, 2018.

Trimmer, M. and Nicholls, J. C.: Production of nitrogen gas via anammox and

denitrification in intact sediment cores along a continental shelf to slope transect in the North Atlantic, Limnol. Oceanogr., 54, 577-589, https://doi.org/10.4319/lo.2009.54.2.0577, 2009.

Trimmer, M., Engström, P., and Thamdrup, B.: Stark Contrast in Denitrification and Anammox across the Deep Norwegian Trench in the Skagerrak, Appl. Environ. Microbiol., 79, 7381-7389, https://doi.org/10.1128/AEM.01970-13, 2013.

Given the fact that we use simple correlation analyses to explore the relationships between nitrogen loss rates and environmental factors, we can't identify which parameter is the most significant controlling factor. However, based on correlation analysis, we identified some key factors influencing denitrification and anammox. We have added these sentences in line 388-393. "Through the correlation analysis of global-scale compiled data, we identified that sediment C/N ratios, oxygen penetration depth, water depth, temperature, salinity, dissolved oxygen, and nitrate concentrations were the main factors regulating denitrification rates, whereas sediment organic carbon, C/N ratios, temperature, salinity, and nitrate concentrations primarily controlled anammox rates (Fig. 5 and Fig. 6)."

In line 453-456, "Based on the simple correlation analysis of global-scale compiled data, we identified that sediment C/N ratios, oxygen penetration depth, water depth and temperature were the primary factors governing the relative contribution of anammox to total nitrogen loss (Fig. 8)."

Line 357 I am wondering about the sediment characteristics at these study sites. Do they include vegetated areas? These factors can significantly influence denitrification and anammox rates.

Response: Thank you for this suggestion. We have checked the references and deleted studies by Gao et al. (2017) and Shan et al. (2016) as their study sites were in vegetated areas. We have restated the expression in line 419-422. "Liu et al. (2020) have examined the spatio-temporal changes of *in situ* nitrogen loss processes in intertidal wetlands of the Yangtze Estuary and found that denitrification was linked to anammox, implying the coupling of denitrification and anammox on a local scale."

Some data in the table represent open ocean environments, while others are from riverine systems. Please consider adding water depth to Table 1 to provide clearer context for the different study sites.

Response: Thank you for this suggestion. We have amended water depth to Table 1.